# Introducing an Innovative Pain Scale for Assessing Postpartum Pain in Mares: Preliminary Clinical Evaluation

**DOI:** 10.3390/ani15233454

**Published:** 2025-11-30

**Authors:** Julia Bolesławska-Szubartowska, Magdalena Kucharczuk, Aleksandra Skrabska, Aneta Zbysław, Julia Adamowicz, Agnieszka Alszko, Klementyna Domagalska-Stomska, Marta Durska, Agata Dziekcierów, Zuzanna Janiszewska, Roksana Korzeniowska, Karolina Kraujutowicz, Karolina Kulesza, Patrycja Marciniak, Zofia Pacyna, Julia Przeborowska, Zuzanna Siwek, Mark Leonard, Anna Rapacz-Leonard

**Affiliations:** 1Students’ Scientific Club ‘Equine Reproduction’, Department of Animal Reproduction with Clinic, Faculty of Veterinary Medicine, University of Warmia and Mazury, Oczapowskiego 14, 10-719 Olsztyn, Poland; 2Leonard Data Lab, 10-719 Olsztyn, Poland

**Keywords:** care, horse, welfare, labor, recovery, soreness, cortisol, saliva, validation

## Abstract

After giving birth, mares can experience significant pain, especially if the birth was difficult. Recognizing and measuring this pain is important for the well-being and recovery of the mare, but until now, there has not been a specific tool to assess it. To address this gap, the researchers developed a pain scale designed to assess pain in mares after they have foaled. The scale uses easy-to-notice signs of discomfort that both veterinarians and horse caretakers can observe. For example, the scale looks at changes in the mare’s behavior and body that might indicate she is experiencing pain. In a study with ten mares that had difficult births, the pain scale was tested and compared with a biological stress indicator: cortisol in the mares’ saliva. Higher levels of this stress hormone usually indicate an animal is feeling more pain or distress. The mares with higher pain scores on the scale also had higher cortisol levels. This shows that the pain scale accurately detects pain. By using this scale, caretakers and veterinarians can quickly identify and treat pain in mares after birth, helping the mares feel better and recover faster.

## 1. Introduction

Pain is a natural defense mechanism that affects both the body and the emotions, and it is commonly associated with tissue injury [1,2]. Pain triggers reflexive changes in respiratory and cardiovascular function and alters levels of hormones (e.g., cortisol, catecholamines, β-endorphins) as well as inflammatory mediators (e.g., prostaglandin E_2_, bradykinin, substance P) [2]. Behaviorally, different species exhibit varying responses to pain: some animals freeze or hide in response to pain, whereas others (such as horses) will attempt to flee from danger (or pain) even when severely injured [1,2].

Mares can experience significant nociceptive pain during the postpartum period, even after normal foaling. This pain mainly results from swelling and irritation of the birth canal and muscle soreness due to the exertion of labor. Importantly, dystocia or postpartum complications such as birth canal injuries, retained fetal membranes, laminitis, mastitis, or post-foaling colic can greatly exacerbate the pain [3]. Accurate assessment and management of pain in these cases are critical for animal welfare and guiding timely and appropriate analgesic interventions. Unrelieved pain can provoke excessive sympathetic nervous system activation, metabolic disturbances, and immunosuppression, which together increase morbidity and impede recovery [4,5]. Because pain intensity cannot be measured directly in animals, veterinarians rely on indirect evaluation methods such as behavioral pain scales. An effective pain scale defines specific behavioral and physiological criteria tailored to the species, enabling the detection of even subtle changes in the animal’s appearance or vital signs that may indicate pain [1,4,5,6]. To ensure the tool is scientifically sound and clinically useful, it must undergo validation with respect to key psychometric properties such as reliability and validity. An important aspect of validation is comparison of the new scale’s scores with an independent indicator of pain, like cortisol levels. Cortisol is a stress hormone known to increase in response to pain in equids [7], as a result of HPA axis activation (CRH→ACTH→adrenal cortex) [8]. Cortisol passively diffuses from blood into the saliva, and salivary cortisol largely mirrors free cortisol with a short lag [9]. Measuring cortisol in saliva is noninvasive because it only requires putting a cotton swab or a Salivette^®^ into a horse’s mouth; therefore, it should not increase the preexisting pain [9].

To the authors’ knowledge, there are no pain scales for postpartum mares. The pain scales currently available for equids include multidimensional composite scales, validated primarily in horses with orthopedic or colic pain [6,10,11,12,13,14,15,16] and in donkeys [17,18,19], as well as facial expression-based scales for adult equids [13,18,20,21,22,23] and for foals [23,24,25]. However, these tools are inadequate for assessing mares after parturition, as they do not specifically capture pain originating from the genital tract. As a result, there is a pressing need for a pain scale tailored specifically for evaluating postpartum mares.

This article aims to describe the creation of a pain scale designed specifically for postpartum mares.

## 2. Materials and Methods

### 2.1. The Development of the Pain Scale for Postpartum Mares

By repurposing pain scales used in orthopedic and abdominal assessments of equids [6,10,11,12,13,14,15,16,17,18,19,20,21,22,23,24,25], as well as those developed for other animal species and non-verbal human infants [26,27,28,29,30,31,32,33,34,35,36,37,38,39,40,41,42,43,44,45], and subjecting them to careful analysis, the authors identified necessary elements for effectively evaluating postpartum mare pain:Observation of the mare in its stall (from a distance): observing posture, position, behavior, and response to surroundings.Observation of the mare’s head: assessing changes in facial expression, ear position, eyelids, nostrils, lips, and tongue.Clinical examination: evaluating respiratory rate and body temperature.Palpation of the mammary gland and body torso: detecting tissue swelling and local temperature changes.Peristalsis examination: assessment of whether feces are being passed and whether their appearance is normal.Hoof examination: checking for local temperature.Food test: evaluating the mare’s reaction to tasty treats.Examination of walking traits: analyzing gait and willingness to walk and turn.

Because the authors also wanted to address the need for breeders to assess the pain of mares, the authors conducted a survey within the Polish online equine community, across Polish horse enthusiast groups on Facebook, to assess which parameters breeders can measure/examine. In the survey, 415 respondents participated: 246 were breeders, 148 were horse owners, 14 were riders, and 7 were veterinarians. On average, the respondents had 6.4 breeding mares in their herds (the survey results are presented in Appendix A).

### 2.2. Animals

Information about the animals that the pain scale was tested on is reported according to ARRIVE guidelines 2.0. The pain scale was tested on n = 10 mares admitted to the Department of Animal Reproduction clinic (Faculty of Veterinary Medicine, University of Warmia and Mazury in Olsztyn, Poland). This was a single-group observational study of mares in the immediate postpartum period after dystocia; no pain-free control group of postpartum mares was included.

All mares suffered from dystocia with dead fetuses (phase two of parturition lasting over 2 h), and all cases of dystocia were treated with fetotomy in a standing position. The mares’ average age was 8.6 years (ranging from 3 to 19 years), and the average weight was 800 kg (ranging from 670 kg to 935 kg). All mares belonged to the Polish heavy draft breed. Written consent was obtained from the owners to evaluate all the mares. The inclusion criteria for testing the pain scale included only postpartum mares; therefore, any other horses admitted to the clinic were excluded from testing. There was no randomization or blinding, as the authors aimed to test the pain scale on the largest possible group of postpartum mares.

### 2.3. Using the Created Pain Scale for Postpartum Mares

After dystocia was resolved, the animals were hospitalized and monitored for 1–4 days, depending on their clinical recovery. Pain was assessed twice daily (at 08:00 and 20:00), at least 30 min after any clinical procedures, feeding, or other handling of the mare. Mares were evaluated in their individual box stalls. For parts 1 and 2 of the pain scale, the observer stood quietly approximately 1–2 m in front of the box door and did not enter the stall or interact with the mare. This part took up to 5 min. The examiner then entered the box to complete parts 3 to 7 of the scale. Finally, the mare was led outside the stall to complete part 8. All scoring was performed in under 30 min. All scores were recorded on printed case-report forms.

Pain assessments were performed by two to three final-year veterinary students (co-authors of this study and members of the Student Scientific Club “Equine Reproduction”). As they had designed the pain scale, the students were familiar with the relevant literature and the principles of pain assessment. In addition, they received standardized training from an experienced equine clinician in the use of the postpartum mare pain scale. Training included a theoretical explanation of each item and its scoring, followed by joint scoring of example cases until agreement was reached on all items. The clinician supervised the students’ assessments and provided feedback to ensure consistent application of the scale. For each mare and time point, two students independently evaluated the mare and scored the scale without discussion. Their scores showed minimal discrepancy (less than 2–4% difference in total scores). When their total scores differed, the students re-evaluated the mare together and agreed on a single consensus score, which was used for analysis.

### 2.4. Cortisol Measurements

At the time of scoring, saliva samples were also taken. Saliva samples were collected using cotton swabs (Regione Monforte, Canelli, Italy) and centrifuged at 15,000 rpm for 15 min. The resulting fluid was then frozen in 2.0 mL Eppendorf tubes at −80 °C. Prior to testing, the saliva samples were thawed and centrifuged for 20 min at 2500 rpm to separate any particulate matter. The centrifuged saliva was then transferred to clean Eppendorf tubes. Each sample was diluted at a ratio of 1:4 using 1x Assay Buffer from the ELISA kit before testing. Each sample was tested in triplicate. The Cortisol Competitive ELISA Kit (one plate, intra-assay coefficient of variation = 8.7%; Invitrogen, Life Technologies corporation, Frederic, MD, USA, Catalog Number EIAHCOR) was used according to the manufacturer’s protocol.

### 2.5. Statistical Methods

Before further analysis, the technical replicates from each mare were screened for outliers. Two points with clearly erroneous values (<22 pg/mL; suspected pipetting errors) were removed, as well as two outliers that were identified with a multivariate projection method [46] implemented with the outpro function in the WRS package, version 40 [46], for R, version 4.4.2 [47]. After outlier removal, there were at least two technical replicates for each time of sampling from each mare.

To quantify the overall association between pain score and salivary cortisol levels while accounting for the correlation between multiple measurements from the same mare, a Bayesian linear mixed-model was used. The model allowed each mare to start at a different cortisol level (random intercepts) and display a different cortisol-per-pain increase (random slopes). A fully-Bayesian approach was chosen over a maximum likelihood method (the traditional approach that can provide *p*-values and confidence intervals) because, with five mares, only the Bayesian approach enabled the inclusion of random slopes for each mare, which reduces false-positive risk [48]. An additional advantage of Bayesian models is that they can incorporate weakly-informative prior information, which further reduces the risk of false positives and enables better out-of-sample predictions by gently regularizing (stabilizing) the results without unduly influencing them [49,50]. Across the observed range of pain scores, the association appeared approximately linear with homoskedastic, normally-distributed residuals; thus, the data were not log-transformed before analysis. A graphical check did not display any evidence that the salivary cortisol levels of the mares followed a circadian rhythm (Appendix A), which is consistent with reports indicating that changes in daily routine or environment alter cortisol levels [51,52] and disrupt cortisol circadian rhythms [53,54]. For these reasons, the model did not include parameters to control for a circadian rhythm.

The prior information for the model was obtained from reports by Schmidt et al. [54] and Bohák et al. [55], which suggested that it was reasonable to expect as much as a four standard deviation (SD) rise in mean cortisol level across the range of observed pain scores (i.e., a one SD rise in cortisol for a one SD increase in pain score). However, to allow for surprises while still regularizing the results, this assumption was weakened. Thus, the prior probability distribution for the model slope coefficient stated that (1) there was an equal probability that an increase in pain score could be associated with an increase or a decrease in cortisol level; (2) changes up to twice as large as those suggested by the results of Schmidt et al. [54] were fairly likely; but (3) smaller changes were more likely than larger ones. Mathematically, this was expressed as a normal distribution centered on zero with an SD of 29.63 pg/mL (equivalent to a 2 SD shift in cortisol for a 1 SD change in pain score; see Figure 1 for a graphical representation). For full details about the other prior distributions, please see the annotated computer code in the Appendix A. Briefly, those prior distributions were set to the defaults in the rstanarm package, version 2.32.1 [50,56], for R, with the modifications to the decomposition of covariance prior suggested by the package developers for use when the number of groups (i.e., mares) is small (https://github.com/stan-dev/stan/wiki/prior-choice-recommendations, retrieved 13 October 2025). These priors are intended to stabilize computation without noticeably influencing the results. A prior predictive check showed that simulating from the combination of these prior distributions tended to produce values that were within the expected order of magnitude (100 s of pg/mL, based on Schmidt et al. [54] and Bohák et al. [55]) (Appendix A), indicating that they would provide mild regularization without having an undue influence on the results.

To sample from the posterior probability distributions, four Markov chains were run for 20,000 iterations each (50% warmup, 50% sampling). There were no divergent transitions, and the resolution and mixing of the chains were excellent, as indicated by Rhat values < 1.001 and bulk effective sample sizes > 12,726 for each parameter, as well as rank plots [57] that displayed nearly uniform distributions (Appendix A, Appendix A). Additionally, a posterior predictive check (Appendix A) showed that the model fit the data well. For these diagnostics, the posterior (version 1.6.1; [58]) and bayesplot (version 1.11.1; [59]) packages for R were used (version 4.5.2).

Additionally, to quantify the effect of a potential leverage point with a pain score of 28, that point was removed, and the model was reevaluated using the prior distributions detailed above. Similarly, to investigate the potential influence of the prior probability distribution for the slope coefficient on the modeling results, the weakly informative prior for the slope was changed to the rstanarm default, which is designed not to have a noticeable influence on the results [50,56], and the model was reevaluated with all data points.

The modeling results are reported below as the median of the posterior draws (i.e., the point estimate of the slope), the probability that the slope is greater than zero (Pr(slope > 0)), and the 95% equal-tailed credible interval (95% CrI, i.e., the range within which the true parameter value is most likely located).

## 3. Results

### 3.1. Pain Scale for Postpartum Mares

The created pain scale focuses on postpartum pain in the genital tract and is simple enough that an average breeder can use it (Table 1). When pain is scored in a postpartum mare, each item on the scale contributes a number of points; thus, more severe pain results in a higher total score. The full scale has a maximum of 47 points when all items are assessed. However, for safety reasons, it is not always possible to perform every component (e.g., in uncooperative or potentially dangerous mares). In such cases, the maximum possible score is adjusted accordingly, and the mare’s pain is expressed as a percentage of this adjusted maximum. For example, if limb and body torso palpation cannot be performed, the maximum score is reduced from 47 to 42 points. A mare scoring 16 out of 42 points would therefore have a pain score corresponding to 38.1% of the adjusted maximum. This approach allows standardized comparison of pain intensity even when some items of the scale are not scored.

On the first day after dystocia, all mares experienced some degree of pain, with scores ranging from 5% to 28% (mean 13%) of the maximum possible score. Three mares were discharged on that first day as the veterinarian in charge decided that their recovery was proceeding well; thus, it was not necessary to prolong their stay in the clinic. Another six mares scored from 3 to 29% (average 12%) of the maximum possible score on the second day, from 6 to 15% (average 9%) on the third day, and from 6 to 9% (average 8%) on the fourth day. One mare developed a postpartum large colon torsion and, on the third day postpartum, was euthanized due to financial constraints on the owner. Her score rose from 12% of the maximum possible score in the morning and 15% in the evening of the first day postpartum to 47% in the morning of the second day. The mare was inaccessible for examination in the evening on the second day.

After careful assessments of the clinical data and considering the mares’ pain tolerance, we established that a score above 40% necessitated analgesic treatment, while mares scoring below 40% experienced minimal to no discomfort.

The pain scale is available free of charge through a web-based application (www.skala.olsztyn.pl) (accessed on 20 November 2025) in both English and Polish. The mobile-optimized website provides real-time feedback on a mare’s pain level. Users may access individual mare profiles by logging in with a Google (Gmail) email address, or they may proceed directly to scoring without logging in. Creating a profile shortens the assessment process by eliminating the need to repeatedly enter the mare’s data and also enables storage of previous results. Users can answer each question by selecting the corresponding option, and a “skip” function is available for circumstances in which a particular assessment element cannot be performed (e.g., when a mare is not accustomed to udder palpation but still requires evaluation). Upon completion of all questions, a summary of the likely pain level is generated. If the pain level is high, the authors strongly recommend consulting a veterinarian for further evaluation and care.

### 3.2. Validation with Salivary Cortisol

The Bayesian linear mixed model indicated that the salivary cortisol level of the mares increased an average of 12.98 pg/mL for a 1 unit increase in pain score (Pr(slope > 0): 98.4%; 95% CrI: 1.38 to 29.65 pg/mL) (Figure 1 and Figure 2). A sensitivity analysis showed that removing a potential leverage point (pain score = 28) increased the slope to 18.39 pg/mL (Pr(slope > 0): 97.3%; 95% CrI: –0.40 to 38.62 pg/mL). Similarly, using a less informative prior increased the slope to 13.21 pg/mL (Pr(slope > 0): 98.4%; 95% CrI: 1.29 to 30.69 pg/mL).

## 4. Discussion

This study introduces a concise, easy-to-apply pain scale for postpartum mares that offers rapid decision support for veterinarians, technicians, owners, and breeders. The scale was designed as a pragmatic clinical tool rather than a purely experimental instrument. In this initial evaluation, higher scores were associated with higher salivary cortisol concentrations under multiple modeling assumptions, supporting biological plausibility and suggesting that the scale captures at least part of the physiological stress response related to pain. In the present dataset, this relationship was well described by an approximately linear association across the observed range of pain scores, and the remaining variation around that line (the residuals) appeared roughly symmetric and bell-shaped. Future studies that include mares with a wider range of pain scores, especially at the upper end of the scale, should formally assess whether a simple linear model remains adequate or whether the relationship between pain score and cortisol becomes curved (nonlinear), for example plateauing or rising more steeply at high scores. Likewise, over a broader range of values, salivary cortisol concentrations may display a more skewed distribution than observed here In such situations, a logarithmic transformation of cortisol or models that assume log-normal rather than normal (Gaussian) residuals are likely to be more appropriate, in line with general recommendations for strictly positive, right-skewed biological data [60].

In contrast to existing multidimensional composite pain scales developed for orthopedic or colic pain in horses and donkeys, and facial-expression scales validated in adult horses and foals [6,10,11,12,13,14,15,16,17,18,19,20,21,22,23,24,25,26], the present tool was tailored specifically to the peripartum context. It deliberately weights criteria most relevant to foaling and the immediate postpartum period, such as udder and perineal changes, and abdominal edema, while also adding hoof monitoring and mares’ posture. Retained fetal membranes and ensuing endotoxemia often lead to laminitis and, in severe cases, can be life-threatening; early recognition of pain and systemic compromise is therefore critical [61,62,63,64]. Unlike some existing composite scales, which focus predominantly on locomotor or peristalsis signs, the postpartum mare scale integrates genital-tract and mammary findings that are unlikely to be highlighted by generic equine pain tools. Based on the results from the online survey (Appendix A), respondents reported very high perceived ability (category 4) for assessing sweating, swelling, fecal output and some behavioral signs, but only very low to low–moderate perceived ability (categories 1–2) for detecting a digital pulse and some temperature-based assessments (hoof, udder). Therefore, the authors developed more illustrative images accompanied by instructions, which were implemented in the pain scale (Table 1).

The scoring system is intended to be both standardized and flexible. When all items can be assessed, the maximum score is 47 points. With uncooperative or potentially dangerous mares, certain components (for example, limb palpation) may be unsafe. In such cases, the maximum attainable score is reduced by the points assigned to the unperformed items, and the mare’s pain burden is expressed as a percentage of this adjusted maximum. This approach allows the scale to be adjusted even for uncooperative mares while preserving comparability of scores across individuals and time points.

Based on the broader equine pain literature [6,10,11,12,13,14,15,16,17,18,19,20,21,22,23,24,25,26], serial scoring is more informative than a single assessment. Regular use of the scale at least once, and ideally twice, daily (e.g., morning and evening) enables monitoring of trajectories rather than isolated values. This is especially important in postpartum mares, where some degree of transient discomfort is expected but persistent or increasing scores may signal complications such as metritis, laminitis and/or systemic inflammation, prompting timely clinical intervention and adjustment of analgesic protocols.

Behavioral indicators remain central to modern equine pain assessment. Changes in facial expression and posture, reactions to palpation of painful areas, and alterations in activity often provide the most sensitive clinical information, particularly because horses may display subtle pain related behaviors [13,22,43,64]. Incorporating such behavioral cues (including features aligned with the Horse Grimace Scale) improves objectivity and consistency among observers [13].

Saliva sampling is a practical, non-invasive alternative to venipuncture and reflects the biologically active, free cortisol fraction; equine validation against ACTH challenge supports its use [65,66]. However, salivary cortisol rises in many non-pain contexts (handling, fear, transport, and exercise); thus, it should be interpreted alongside clinical findings and behavior [67,68,69]. Even so, when available as part of a structured protocol, salivary cortisol remains among the best non-invasive physiological anchors for scale validation in horses. In acute abdominal disease, for example, salivary cortisol serves as a useful biomarker of pain-related stress [70].

From a practical perspective, the scale’s clarity and brevity are intended to reduce rater dependence. Use by trained lay caregivers (e.g., breeders or farm staff) could standardize monitoring between veterinary visits, prompt earlier contact with a veterinarian in deteriorating cases, and provide clinicians with structured information to triage cases and adjust therapy. Wider adoption via a mobile-friendly website could elevate routine postpartum surveillance and welfare by supporting serial assessments and response-to-treatment tracking.

Several limitations must be acknowledged, and together they justify describing the present work as a preliminary validation. First, the sample size was modest and derived from a single clinical population of Polish heavy draft mares with dystocia treated by fetotomy. This may limit generalizability to mares after uncomplicated foaling, to other breeds, and to different dystocia treatments. Broader multicenter, prospective studies will be needed to test this scale on a larger population of mares, comprising different breeds, with and without dystocia, and subjected to different dystocia treatments.

The second limitation of this study is the absence of a pain-free control group of healthy postpartum mares. Consequently, we could not formally define the range of behaviors and physical findings compatible with “normal” postpartum discomfort, nor establish cut-off values discriminating physiological post-foaling changes. We were also unable to fully assess the specificity of the scale for pain as opposed to other sources of stress, since the evaluated mares were patients of the clinic, which was already stressful for them. In practice, defining truly “pain-free” postpartum mares is challenging, because some degree of tissue trauma and uterine involution-related discomfort is expected even after uncomplicated foaling. Future validation work should therefore aim to include comparison groups, such as mares after uncomplicated parturition, to better characterize specificity and thresholds.

Another limitation is that formal inter- and intra-rater reliability could not be evaluated. In this study, pain scores were recorded as consensus ratings of two trained students, and individual ratings were not stored in the database. Although the observers underwent training, practiced on example cases and were supervised by an experienced equine clinician, this approach precludes retrospective calculation of intraclass correlation coefficients or kappa statistics.

## 5. Conclusions

We present a simple, preliminarily validated, postpartum-specific pain scale for mares that combines clinical and behavioral indicators and is available online for serial use by clinicians and caregivers. In this initial study, pain scores increased in parallel with salivary cortisol concentrations, supporting the biological plausibility of the scale. However, the small sample size and the non-specific nature of cortisol as a stress biomarker are important limitations, and external validation with larger and more diverse samples is required. When used routinely, this scale has the potential to standardize pain detection, guide timely analgesic and supportive interventions, and thereby improve postpartum welfare and recovery in mares.

## Figures and Tables

**Figure 1 animals-15-03454-f001:**
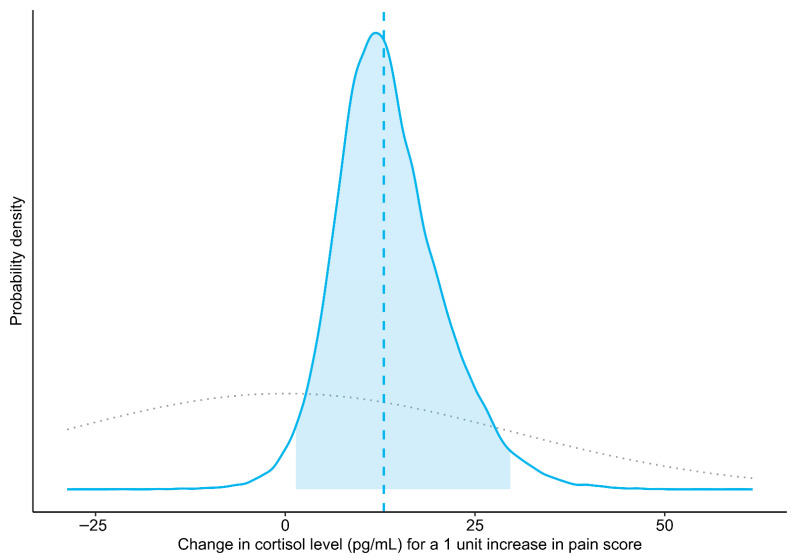
Prior (dotted gray line) and posterior probability distributions (solid blue line) for the mean change in salivary cortisol level associated with a one-point increase in pain score (slope of the Bayesian linear mixed model). The vertical dashed line indicates the point estimate of the slope (median of posterior draws), and the shaded area represents the 95% credible interval (central 95% of the posterior distribution).

**Figure 2 animals-15-03454-f002:**
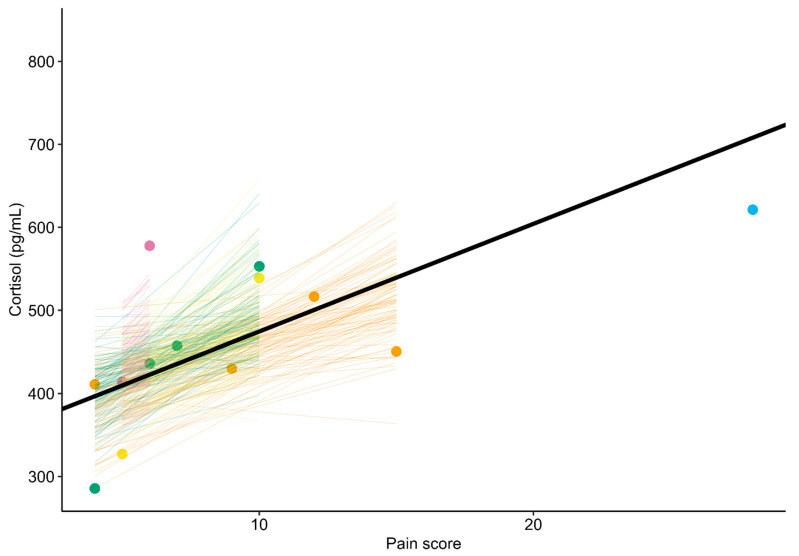
Association between pain score and salivary cortisol level in five postpartum mares. The thick black line represents the overall trend estimated by the Bayesian linear mixed model. Colors correspond to individual mares. Points show observed pain scores and associated cortisol levels. Colored lines illustrate the uncertainty in the modeled trends for each mare (each line corresponds to one of 100 random draws from the expectation of the posterior predictive distribution).

**Table 1 animals-15-03454-t001:** Postpartum mare pain-scale for use by veterinarians and caretakers (comprising eight assessment components).

PAIN SCALE FOR HEAVY MARES AFTER DELIVERY
Date and time:
Mare’s name:
Breed:
Age:
Weight:
PART 1. Observation of a mare in a stall (gray cells do not score points)
Points to score	0	1	2	3	4	Score
**Posture and weight distribution**	Normal, symmetrical weight bearing on all limbs	Occasionally shifting weight off one limb (mild, intermittent favoring)	Abnormal stance without clear favoring of a specific limb	Abnormal stance with obvious favoring of one or more limbs or frequent weight shifting	Marked and continuous weight shifting between limbs; clear reluctance to bear weight on one or more limbs	
**Position of the** **mare’s body**	Standing freely with a normal, upright posture	Standing but with obvious muscle tension	Occasional attempts to lie down	Frequently lying down and getting up	Lying down all the time, with visible attempts to roll or wallow	
**Position and movement of the mare’s tail**	Tail in a normal position with free, calm movements	Elevated tail, with minimal/no movements				
**Mare’s interest in her abdomen/vulvar area**	No particular interest in the abdomen or vulvar area		Occasionally looking at her flank/vulvar area		Marked interest; kicks abdomen with hind limbs	
**Vocalization**	No pain related vocalization	Panting/groaning				
**Mare’s reaction to surroundings and sounds**	Interested in surroundings, easy to approach and engage		Observes her surroundings, but reduced response to sounds		Little or no interest in surroundings, minimal or absent response to sounds	
**Sweating**	No visible sweating	Visible sweating				
**Other/notes:**		**Summary:**	**/19**
**PART 2. Observation of the mare’s facial expression-Horse Grimace Scale (gray cells do not score points)**
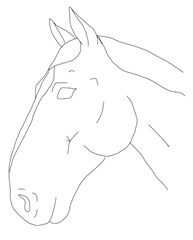	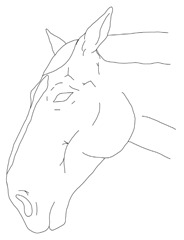	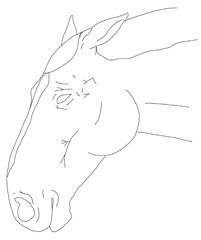
a. Diagram of expression equivalent to 0 points on the scale	b. Diagram of expression equivalent to 1 point on the scale	c. Diagram of expression equivalent to 2 points on the scale
**Points to score**	**0**	**1**	**2**	**Score**
**Visibility of wrinkles on the head**	Wrinkles not visible, relaxed facial expression	Wrinkles visible	Pronounced, clearly visible wrinkles	
**Ear position**	Standing up	Slightly laid back	Completely laid back	
**Eyelids**	Open	Squinting	Marked squinting, eyelid covers approximately half of the eyeball	
**Nostrils**	Relaxed	Abnormally wide	Extremely wide	
**Turning back the lips/teeth gnashing**	No		Yes	
**Tongue protruding**	No		Yes	
**Other/notes:**		**Summary:**	**/12**
**PART 3. Clinical examination**
Instructions for measuring core temperature: Place an electronic thermometer in the mare’s rectum and turn it on. Read the result when the signal is heard indicating that the thermometer is ready.
Instructions for measuring respiratory rate: This examination should be performed when the mare is standing quietly and has not recently undergone any physical effort. The examiner’s palm should be placed close to the mare’s nostril. The respiratory rate is the number of exhalations counted over one minute.
**Points to score**	**0**	**1**	**2**	**Score**
**Respiratory rate** 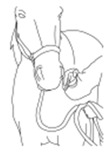	8–16 breaths/min	17–30 breaths/min	>30 breaths/min	
**Body temperature** 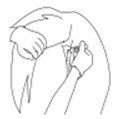	37.0–38.4 °C	36.0–36.9 °C/38.5–39.5 °C	<36 °C or >39.5 °C	
**Others/notes:**		**Summary:**	**/4**
**PART 4: Palpation of the mammary gland and body torso**
*(Presence of swelling and/or areas of increased temperature)*
**Points to score**	**0**	**1**	**Score**
**Back, loin and croup** 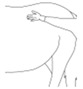	No swelling or increased warmth detected	Swelling and/or increased warmth present	
**Abdomen and flank** 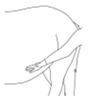	No swelling or increased warmth detected	Swelling and/or increased warmth present	
**Udder** 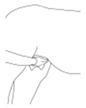	No swelling or increased warmth detected	Swelling and/or increased warmth present	
**Other/notes**		**Summary:**	**/3**
**PART 5: Peristalsis examination**
**Points to score**	**0**	**1**	**Score**
**Fecal output**	Normal consistency and regular defecation	Abnormal fecal consistency and/or no feces passed	
**Others/notes**		**Summary:**	**/1**
**PART 6: Hoof examination**
**Side**	**Hoof**	**Points to score:** **0**	**Points to score:** **0.5**	**Score**
**Left**	Front	Normal hoof temperature similar to other limbs	Increased hoof temperature (noticeably warmer than other limbs)	
Back	Normal hoof temperature similar to other limbs	Increased hoof temperature (noticeably warmer than other limbs)	
**Right**	Front	Normal hoof temperature similar to other limbs	Increased hoof temperature (noticeably warmer than other limbs)	
Back	Normal hoof temperature similar to other limbs	Increased hoof temperature (noticeably warmer than other limbs)	
**Others/notes** 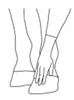		**Summary:**	**/2**
**PART 7: Food test**
**Points to score**	**0**	**1**	**2**	**Score**
**Reaction to tasty treats**	Reaches for food, clearly interested; chews normally	Looks at food and shows some interest; attempts to eat but has visible difficulty chewing (feed falls from mouth, abnormal mandible movements)	Complete lack of interest in food and/or bits of food falling out through the nostrils	
**Others/notes**		**Summary:**	**/2**
**PART 8: Examination of walking traits**
**Points to score**	**0**	**2**	**4**	**Score**
**Behavior in motion**	Walks normally with typical gait and behavior	Abnormal gait (e.g., stiffness, reluctance to move, balking, or lameness)	Unable or unwilling to stand and walk; remains lying down and cannot be encouraged to rise	
**Other/notes:**		**Summary:**	**/4**
**GENERAL TOTAL SCORE: ………/…..…**
**RESULT IN % (of maximum possible score): …………………**

## Data Availability

The anonymized data together with the R script are available in Zenodo at DOI: https://doi.org/10.5281/zenodo.17450756.

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
