# Peer review of "Introducing an Innovative Pain Scale for Assessing Postpartum Pain in Mares: Preliminary Clinical Evaluation"

_animals, 2025, doi:10.3390/ani15233454_

Round 1

Reviewer 1 Report

Comments and Suggestions for Authors

The authors describe a pain scale for the post-partum mare. I would like to express some concerns regarding the study. In my opinion, the most significant limitation is the absence of testing the pain scale on a control group that is definitively free of pain. Furthermore, the authors do not present their work as preliminary or exploratory, which would have been appropriate given the current findings. In my opinion, statistical analysis lacks consideration of the scale’s internal consistency. Additionally, the discussion section is rather brief and does not address these critical limitations. These aspects should be carefully considered to strengthen the validity of the pain scale. Furthermore, I would suggest to improve the layout of the pain scale and of the website (in the website in some item of the scale, the title of the item disappear-see lines 250-252). I suggest also more clarity in the M&m section (see below).

Specific points:

Simple summary and abstract

It should be simple but not use inappropriate terms. My suggestions:

Line 16: after giving birth, mares can…

Line 19: …researchers developed a pain scale designed for mares…

Line 23: remove “NEW” throughout the abstract

Line 24: the team measured a hormone… replace with “Cortisol in the mare’s saliva was measured;..”

Introduction

Line 89: Orthopedic and colic patients, this is incorrect because there are also foals. And  actually in that parenthesis, the best distinction that can perhaps be made is between composite scales and facial expression scales. Furthermore this reference is missing, which is essentially the validation of the facial scale for foals. van Loon, J. P., Trindade, P. H. E., da Silva, G. V., Keus, J., Huberts, C., de Grauw, J. C., & Lanci, A. (2025). Objective assessment of acute pain in foals using a facial expression‐based pain scale. Equine Veterinary Journal.

M&m

It is not very clear how a user should evaluate the mare. From inside the box, from a distance, for how long? In my opinion, a clearer description is needed. The students assess the mares, but are you sure they assess them properly? And how did you evaluate their scores? Do you consider the average of those given by the two students?

Minor comments

Line 97-98: Please remove “To the authors’ knowledge, there are no pain scales for postpartum mares. To address this gap”. This concept is part of the introduction.

Line 98-101: Please combine these two sentences into one.

Line 108: and the evaluation of a reaction of the mare?

Line 122: add the University of the department

Line 122-124: remove the comment about the control group. Just say that there is no control group. Although, in my opinion, this remains the biggest concern, the fact that it has not been tested on pain-free mares. So, honestly, I would refer to them as preliminary results, even in the title.

Line 124: How was the dystocia resolved? Was the same method used in all 10 cases? I think it may vary depending on whether AVD, CVD fetotomy, or C-section was performed.

Line 128-129: Are the inclusion criteria simply being a postpartum mare?

Statistical analysis

The Bayesian linear mixed model was used to analyze the association between pain scores and salivary cortisol levels while accounting for repeated measurements within mares. The model included random intercepts and slopes to capture individual variability in baseline cortisol and response to pain. The Bayesian approach was chosen to stabilize parameter estimates with a small sample size and to incorporate weakly informative priors, reducing the risk of false positives. Model diagnostics indicated good fit and stable inference, supporting its suitability for this validation study. However, for the validation of a new pain scale, I think it is important to perform an internal consistency analysis (e.g., using Cronbach’s alpha) to assess the coherence among the items of the scale. Additionally, it is important to evaluate both inter-rater and intra-rater reliability to verify the reproducibility of the scale between different observers and across repeated assessments by the same observer. These tests together ensure that the scale is both internally consistent and reliable over time and between different users. I would say that, at the very least, the authors should consider making the manuscript “preliminary”, adding an appropriate paragraph for discussion, which is, however, far too short.

Results

This paragraph does not seem very clear or well organised to me. I suggest revising it and removing comments that could be discussed in the discussion section.

Line 215-I suggest changing in “The created pain scale focuses on postpartum pain from the genital tract, but…”

Lines 217-226 The scoring system is not immediately clear from these sentences. I recommend rephrasing them in a simpler and more logical way.

Line 227-9: this is a comment. I suggest delating this sentence (-> discussion).

Line 230-1: I suggest rephrasing it, removing ‘that the scale was tested on experienced some pain’

Line 231: three mares were discharged.. that day?

Line 233-4. Remove “the owners were advised …….consultation”

Line 237: was euthanized due to financial constraints on the…

Discussion &References

Too short. Several other points could be touched upon, such as the discussion of control groups, comparison with other composite scales, etc.

The formatting of the references needs to be fixed: many articles do not have the year where it should be and italics and bold where they should be.

Comments on the Quality of English Language

In some places, the English in particular could be improved.

Author Response

We are very grateful to Reviewer 1 for the thorough evaluation of our manuscript and the many constructive comments and suggestions. We have carefully revised the manuscript in light of these remarks. Below we respond point-by-point to each comment and indicate how the manuscript has been modified. All changes are highlighted in yellow in the revised version.

Comment 1:
“The authors describe a pain scale for the post-partum mare. I would like to express some concerns regarding the study. In my opinion, the most significant limitation is the absence of testing the pain scale on a control group that is definitively free of pain. Furthermore, the authors do not present their work as preliminary or exploratory, which would have been appropriate given the current findings. In my opinion, statistical analysis lacks consideration of the scale’s internal consistency. Additionally, the discussion section is rather brief and does not address these critical limitations. These aspects should be carefully considered to strengthen the validity of the pain scale. Furthermore, I would suggest to improve the layout of the pain scale and of the website (in the website in some item of the scale, the title of the item disappear-see lines 250-252). I suggest also more clarity in the M&m section (see below).”

Response 1:
We sincerely thank the reviewer for these important comments. In response, we have implemented the following changes:

  • Control group and preliminary nature. We now explicitly state in the Methods that this was a single-group observational study of mares with postpartum pain after dystocia and that no pain-free control group was included. We have also revised the title and text to present the work clearly as a preliminary clinical evaluation. The preliminary nature of the findings and the absence of a pain-free control group are now emphasised as major limitations in both the Abstract and the Discussion.
  • Internal consistency and reliability. We fully agree that internal consistency and reliability are key aspects of pain scale validation. Unfortunately, only total scores were entered into our database and item-level data are no longer available. It is therefore not possible to retrospectively compute indices such as Cronbach’s alpha. Likewise, individual observers’ ratings were not stored, so formal inter- and intra-rater reliability statistics (e.g., ICC, kappa) cannot be calculated for this dataset. We now explicitly acknowledge these points as important limitations and state that a full psychometric evaluation, including internal consistency and reliability, must be undertaken in future validation studies.
  • Discussion. The Discussion has been substantially expanded. It now includes a broader consideration of control groups, a more detailed discussion of limitations, an explanation of why our model was the best choice for analyzing this dataset, and future directions. We also emphasise that the present work represents a preliminary validation based on a small clinical sample.
  • Layout of pain scale and website. The layout of the pain scale tables in the manuscript has been revised so that all items and headings are clearly visible and consistently formatted. Regarding the website, we have re-checked the content and confirmed that the item titles display correctly on standard desktop, laptop, and phone screens. We note that on some smaller devices, users may need to scroll to view all content, which may have contributed to the reviewer’s experience. At present, the website is a student-led resource created without external funding, and while we cannot guarantee optimal display on every device, we will continue to improve its usability as far as our technical and financial resources allow.
  • Materials and Methods clarity. As detailed in later responses, we have clarified the pain assessment procedure (observer position, duration, and scoring) and the training and scoring protocol for the student observers.

Simple Summary and Abstract

Comment 2 (Line 16):
“It should be simple but not use inappropriate terms. My suggestions: Line 16: after giving birth, mares can…”

Response 2:
We thank the reviewer for this suggestion. The sentence in the Simple Summary around line 16 has been revised as recommended, using the wording “after giving birth, mares can…”.

Comment 3 (Line 19):
“Line 19: …researchers developed a pain scale designed for mares…”

Response 3:
We agree. The phrase has been revised to “researchers developed a pain scale designed for mares…”.

Comment 4 (Line 23):
“Line 23: remove ‘NEW’ throughout the abstract.”

Response 4:
We have removed the word “NEW” wherever it appeared.

Comment 5 (Line 24):
“Line 24: the team measured a hormone… replace with ‘Cortisol in the mare’s saliva was measured;..’”

Response 5:
We appreciate this clarification. The sentence has been revised to: “Cortisol in the mare’s saliva was measured; …”.

Introduction

Comment 6 (Line 89):
“Orthopedic and colic patients, this is incorrect because there are also foals. And actually in that parenthesis, the best distinction that can perhaps be made is between composite scales and facial expression scales. Furthermore this reference is missing, which is essentially the validation of the facial scale for foals. van Loon, J. P., Trindade, P. H. E., da Silva, G. V., Keus, J., Huberts, C., de Grauw, J. C., & Lanci, A. (2025). Objective assessment of acute pain in foals using a facial expression‐based pain scale. Equine Veterinary Journal.”

Response 6:
We thank the reviewer for this important clarification. The relevant sentence has been revised to distinguish explicitly between multidimensional composite pain scales and facial expression-based scales and to mention that such scales have been also validated in donkeys and foals. We have also added the suggested reference (van Loon et al., 2025).

Materials and Methods

Comment 7:
“It is not very clear how a user should evaluate the mare. From inside the box, from a distance, for how long? In my opinion, a clearer description is needed. The students assess the mares, but are you sure they assess them properly? And how did you evaluate their scores? Do you consider the average of those given by the two students?”

Response 7:
We appreciate these detailed and practical comments. The pain assessment procedure and observer protocol have been clarified as follows:

  • We now specify that mares were evaluated in their individual box stalls. For the initial parts of the scale, the observer stood quietly approximately 1–2 m in front of the box door, did not enter the stall, and did not interact with the mare. The mare was observed for a standardised period, and subsequent parts of the scale were completed after entering the box or, for the final part, leading the mare outside.
  • We have expanded the description of observer training and standardisation. Two to three final-year veterinary students (co-authors, members of the Students’ Scientific Club “Equine Reproduction”) performed the assessments. They had designed the scale and were familiar with the relevant literature, and they also received structured training from an experienced equine clinician, including theoretical explanation of each item, joint scoring of example cases, and supervised early assessments with feedback.
  • For each mare and time point, two students scored the scale independently and without discussion. Their scores generally differed by less than 2–4% of the total score. Discrepancies were resolved by joint re-evaluation and a consensus score, which was used for analysis. We have clarified in the Methods that consensus scores were recorded and that individual ratings were not stored, which is also acknowledged as a limitation in the Discussion.

These details now appear in the Materials and Methods section under the “2.3 Using the created pain scale for postpartum mares”

Minor Comments

Comment 8 (Lines 97–98):
“Please remove ‘To the authors’ knowledge, there are no pain scales for postpartum mares. To address this gap’. This concept is part of the introduction.”

Response 8:
We agree that this wording was redundant. The phrase has been removed.

Comment 9 (Lines 98–101):
“Please combine these two sentences into one.”

Response 9:
We have combined the two sentences into a single, clearer sentence as suggested.

Comment 10 (Line 108):
“Line 108: and the evaluation of a reaction of the mare?”

Response 10:
We thank the reviewer for drawing attention to this point. This item is not considered valid for evaluating the mammary gland. Many mares are not accustomed to udder palpation and may show distress simply in response to touch. Therefore, in our scale, only swelling and increased local temperature are considered indicative of inflammation (and thus pain) in the mammary gland. The text has been revised accordingly (Table 1, part 4).

Comment 11 (Line 122):
“Add the University of the department.”

Response 11:
We have added the name of the University to the departmental affiliation at the indicated lines.

Comment 12 (Lines 122–124):
“Remove the comment about the control group. Just say that there is no control group. Although, in my opinion, this remains the biggest concern, the fact that it has not been tested on pain-free mares. So, honestly, I would refer to them as preliminary results, even in the title.”

Response 12:
We thank the reviewer for this crucial clarification. We have removed the previous wording about the control group and replaced it with the sentence:

“This was a single-group observational study of mares in the immediate postpartum period after dystocia; no pain-free control group of postpartum mares was included.”

We have also modified the title to “Introducing an Innovative Pain Scale for Assessing Postpartum Pain in Mares: Preliminary Clinical Evaluation” and explicitly refer to the findings as preliminary in the Abstract and Discussion. This clearer wording, together with the expanded limitations, more accurately reflects the current level of evidence and the need for future validation in pain-free postpartum mares.

Comment 13 (Line 124):
“How was the dystocia resolved? Was the same method used in all 10 cases? I think it may vary depending on whether AVD, CVD fetotomy, or C-section was performed.”

Response 13:
We appreciate this point. All cases in our study were treated using the same method: fetotomy in a standing position. We have clarified this in the Animals section by adding the sentence:

“…and all cases of dystocia were treated with fetotomy in a standing position.”

Comment 14 (Lines 128–129):
“Are the inclusion criteria simply being a postpartum mare?”

Response 14:
We agree that the inclusion criteria needed to be specified more clearly. The only inclusion criterion was that the mare was in the immediate postpartum period after dystocia. We did not include a separate pain-free control group of postpartum mares. As mentioned earlier, defining a truly “pain-free” postpartum mare is challenging in clinical practice, because some degree of discomfort is expected even after uncomplicated foaling. We now state explicitly in the Methods that no control group was included and emphasise this as a key limitation in the Discussion.

Statistical Analysis

Comment 15:
“The Bayesian linear mixed model was used… However, for the validation of a new pain scale, I think it is important to perform an internal consistency analysis (e.g., using Cronbach’s alpha)… Additionally, it is important to evaluate both inter-rater and intra-rater reliability… These tests together ensure that the scale is both internally consistent and reliable over time and between different users. I would say that, at the very least, the authors should consider making the manuscript ‘preliminary’, adding an appropriate paragraph for discussion, which is, however, far too short.”

Response 15:
We thank the reviewer for these very pertinent comments. Our responses are as follows:

  • Internal consistency. We fully agree that internal consistency should be assessed. Unfortunately, only the total scores of the scale were entered into the database and item-level scores were not retained. We are therefore unable to retrospectively compute Cronbach’s alpha or similar indices for this dataset. We now explicitly state this as a limitation.
  • Inter- and intra-rater reliability. We also agree on the importance of inter- and intra-rater reliability. As noted earlier, pain scores were recorded as consensus ratings of two trained observers, and individual ratings were not stored. Consequently, it is not possible to retrospectively calculate inter- or intra-rater reliability statistics (e.g., ICC or kappa). We have now explicitly acknowledged this as an important limitation in the Discussion and highlighted the need for future studies specifically designed to collect separate and repeated ratings for reliability analysis.
  • Preliminary nature and expanded Discussion. In accordance with the reviewer’s suggestion, we now clearly present the study as a preliminary clinical evaluation. The title has been changed to “Introducing an Innovative Pain Scale for Assessing Postpartum Pain in Mares: Preliminary Clinical Evaluation”, and both the Abstract and Discussion emphasise that the results represent preliminary validation based on a small sample, without a pain-free control group and without formal reliability testing. The Discussion section has been expanded to address these methodological limitations and to outline priorities for future validation work.

Results

Comment 16:
“This paragraph does not seem very clear or well organised to me. I suggest revising it and removing comments that could be discussed in the discussion section.”

Response 16:
We thank the reviewer for this observation and have revised the relevant paragraph in the Results to improve clarity and organisation, while moving interpretative comments to the Discussion. In particular:

  • The sentence beginning “The created pain scale focuses on postpartum pain from the genital tract, but…” has been re-phrased as suggested.
  • The description of the scoring system (lines 217–226) has been re-written in a simpler and more logical way, as requested.
  • The sentence noted as a comment at lines 227–229 has been removed from the Results and, where appropriate, incorporated into the Discussion.

Comment 17 (Line 230–231):
“I suggest rephrasing it, removing ‘that the scale was tested on experienced some pain’.”

Response 17:
We agree and have rephrased the sentence to:

“On the first day after dystocia, all mares experienced some degree of pain, with scores ranging from 5% to 28% (mean 13%) of the maximum possible score.”

Comment 18 (Line 231):
“Three mares were discharged.. that day?”

Response 18:
Yes, this referred to the same first day. To clarify, we have revised the text to state that three mares were discharged “on that first day”.

Comment 19 (Lines 233–234):
“Remove ‘the owners were advised …….consultation’.”

Response 19:
We have removed this sentence from the Results section, as suggested.

Comment 20 (Line 237):
“was euthanized due to financial constraints on the…”

Response 20:
We have corrected the sentence to read:

“was euthanized due to financial constraints on the part of the owner.”

Discussion and References

Comment 21:
“Too short. Several other points could be touched upon, such as the discussion of control groups, comparison with other composite scales, etc.”

Response 21:
We agree that the Discussion was too brief. It has now been substantially expanded to:

  • Discuss more fully the absence of a pain-free control group.
  • Address the preliminary nature of the findings and outline priorities for further validation (including larger samples, control groups, and reliability testing).
  • Explanation of why, with current data, our model was the best choice, and what future studies should take into account with a bigger and maybe broader set of data.

Comment 22:
“The formatting of the references needs to be fixed: many articles do not have the year where it should be and italics and bold where they should be.”

Response 22:
We have carefully revised the reference list to conform to the Animals (MDPI) style. Journal titles are italicised, years and volumes are placed in the correct order, and formatting (including italics and punctuation) has been standardised throughout.

English Language

Comment 23:
“In some places, the English in particular could be improved.”

Response 23:
We have improved the English throughout the manuscript. The revised version has been reviewed by a native speaker (coauthor - MEL), a graduate of Emory University in the USA, who helped us refine grammar, phrasing, and clarity.

Reviewer 2 Report

Comments and Suggestions for Authors

Honestly, the article is impressive, well-documented (71 references studied and used in documentary research), with related activities - online forms/website (web developing), and of course the utility is more than obvious - to improve the welfare level of these wonderful animals.
However, there are some minor corrections that can really improve the quality of the content and the visibility of the article:
- Do not repeat the content. Table 1 is almost the same as Table S3. Also, you can create a clear distinction of the 8 directions (parts) in the sub-headings of the table (add Part 1 - ...., Part 2 - .... a.s.o.). Use in the table "respiratory rate" instead of RR. 
- Include the pain scale use at Methods as a separate sub-chapter;
- Add welfare among the keywords;
- Some minor re-wording are required ("she is hurting" is too much, "as they overlook pain within the genital tract and sometimes nowhere else" is unclear and not the meaning you wanted there, use "peristalsis", not peristaltic everywhere, "wallowing", not "vallowing", maybe "signs", not symptoms etc.) and I don't completely agree with the term trunk, try to replace it;
- For the section at lines 230-243 specify that the percentages are out of the maximum possible score;
- Table S1 with no title;

- At line 437 give some details (maybe the project ID or approving no. - you only gave the framework).

Please, follow the detailed comments/suggestions in the attached .pdf file, in order to improve the value and the scientific soundness of your manuscript.
And congrats, we need articles like this for most of the animal species!

Author Response

We would like to thank Reviewer 2 for the careful reading of our manuscript and the positive overall assessment. We are grateful for the constructive suggestions, which have helped us to improve the clarity, structure, and presentation of the work. Below, we provide a point-by-point response to each comment. All corresponding revisions have been made in the manuscript and are highlighted in yellow.

Comment 1:
“Do not repeat the content. Table 1 is almost the same as Table S3. Also, you can create a clear distinction of the 8 directions (parts) in the sub-headings of the table (add Part 1 - ...., Part 2 - .... a.s.o.). Use in the table ‘respiratory rate’ instead of RR.”

Response 1:
Thank you for this helpful comment. We agree that the previous presentation was redundant and that the parts of the scale needed clearer separation. In the revised manuscript, we have removed Table S3 from the Supplementary Materials to avoid duplication with Table 1. In addition, we have edited Table 1 to clearly indicate the eight parts of the scale by adding sub-headings “Part 1–8” to the relevant sections. We have also replaced the abbreviation “RR” with the full term “respiratory rate” (pease see revised Table 1 in the manuscript).

Comment 2:
“Include the pain scale use at Methods as a separate sub-chapter;”

Response 2:
We appreciate this suggestion. We have now added a separate subsection in the Materials and Methods specifically describing the use of the pain scale, under the heading “2.3 Using the created pain scale for postpartum mare””. This subsection explains the assessment procedure, timing, and scoring in a more structured and visible way.

Comment 3:
“Add welfare among the keywords;”

Response 3:
Thank you for this comment. We have added “welfare” to the list of keywords to better reflect the welfare-oriented aims and implications of the study.

Comment 4:
“Some minor re-wording are required (‘she is hurting’ is too much, ‘as they overlook pain within the genital tract and sometimes nowhere else’ is unclear and not the meaning you wanted there, use ‘peristalsis’, not peristaltic everywhere, ‘wallowing’, not ‘vallowing’, maybe ‘signs’, not symptoms etc.) and I don't completely agree with the term trunk, try to replace it;”

Response 4:
We thank the reviewer for these detailed language suggestions. We have carefully revised the wording throughout the manuscript to improve precision and tone. Specifically:

  • Expressions such as “she is hurting” have been replaced with more neutral, scientific formulations (e.g., “she is experiencing pain”).
  • The sentence describing existing pain scales has been rewritten to clarify that they overlook pain originating from the genital tract, which was our intended meaning.
  • The term “peristalsis” is now used consistently instead of “peristaltic” when referring to intestinal motility.
  • Spelling errors (e.g., “vallowing”) have been corrected (e.g., to “wallowing,” where appropriate).
  • We have replaced “symptoms” with “expression”, according to the suggestion from the PDF file.
  • In response to the reviewer’s concern about the term “trunk,” we have replaced “trunk” with “body torso” throughout the manuscript to align better with equine anatomical terminology.

Comment 5:
“For the section at lines 230-243 specify that the percentages are out of the maximum possible score;”

Response 5:
We agree that this section required clarification. We have completely revised the relevant paragraph to make explicit that the reported percentages are calculated relative to the maximum possible score under the conditions of each assessment. The revised text specifies that, when not all items can be scored (for example, in uncooperative mares), the maximum possible score is reduced accordingly, and the mare’s pain is expressed as a percentage of this adjusted maximum.

Comment 6:
“Table S1 with no title;”

Response 6:
Thank you for pointing this out. In our original submission, Table S1 did have a title, but it may not have appeared clearly in the generated PDF (it was combined with the titles of all supplementary materials).

Comment 7:
“At line 437 give some details (maybe the project ID or approving no. - you only gave the framework).”

Response 7:
We appreciate this suggestion to provide more precise information on funding. We have expanded the Funding section to include the requested details. The relevant paragraph now reads:

Two sources of funding were obtained. The first was awarded to a scientific club from Regional Initiative of Excellence Program in the veterinary discipline. This funding was granted for their project titled "Application of a Scale for Assessing Postpartum Pain in Polish Heavy-Draft Mares After Dystocia," totaling PLN 4,000.00 (RID DECISION 3/8/2024/WMW). 

The second source of funding was allocated to cover the Article Processing Charge (APC) for this article, again funded by the Minister of Science under „the Regional Initiative of Excellence Program" (no ID number).

Comment 8:
“Please, follow the detailed comments/suggestions in the attached .pdf file, in order to improve the value and the scientific soundness of your manuscript.”

Response 8:
We thank the reviewer for the careful, line-by-line reading of the manuscript. We have gone through all detailed comments and suggestions in the annotated PDF and have implemented the recommended corrections and clarifications throughout the manuscript. All such changes are visible in yellow in the revised version.

Comment 9:
“And congrats, we need articles like this for most of the animal species!”

Response 9:
We sincerely thank the reviewer for this encouraging remark!

Round 2

Reviewer 1 Report

Comments and Suggestions for Authors

Congratulations! Good job! Now, in my opinion, the manuscript has really improved. I am very happy and proud.